# Roles of Agricultural Cooperatives (ACs) in Drought Risk Management among Smallholder Farmers in Pursat and Kampong Speu Provinces, Cambodia

**Nyda Chhinh [1,\*], Serey Sok [2], Veasna Sou [2] and Pheakdey Nguonphan [3]**

1    Department of Economic Development, Royal University of Phnom Penh, Phnom Penh 12150, Cambodia
2    Research Office, Royal University of Phnom Penh, Phnom Penh 12150, Cambodia
3    Centre for Culture and Peace Studies (CCPS), Asian Vision Institute (AVI), Phnom Penh 12150, Cambodia
\*    Correspondence: chhinh.nyda@rupp.edu.kh; Tel.: +855-1295-5169

**Abstract:** This research aims to investigate the roles of agricultural cooperatives (ACs) in the provinces Pursat and Kampong Speu, Cambodia, with respect to managing the drought risks among smallholder farmers, with particular focus on the following factors: (a) the impacts of drought on socio-economic development and livelihood; (b) the services delivered by AC operations in terms of increasing the five livelihood assets; and (c) the interactions between AC operations, adaptive capacity, and the impacts of drought. Household surveys were conducted among 421 smallholder farmers; in addition, case studies were also conducted with stakeholders in the Bakan district in Pursat Province and the Barsedth district in Kampong Speu Province. The study demonstrates that (i) both climatic and human-made factors contributed toward the impact of drought in the Bakan and Barsedth districts. Furthermore, this hazard affected smallholder farmers. (ii) AC operations increased the smallholder farmers' access to natural and physical assets. Currently, AC operations are constrained by a lack of willingness and commitment in AC committees, trust building in the communities, and human and financial resources after development projects, which are implemented by non-governmental organizations (NGOs). (iii) Adaptive capacity contributed to the drought impacts and participation in AC activities, but the involvement in AC activities did not contribute to AC operations nor to aiding with the impacts of drought. This empirical research, which was performed via structural education modeling (SEM), fills a gap in the literature by increasing the understanding of the roles of AC operations in the context of drought risk management and their role in increasing access to the five livelihood assets.

**Keywords:** agricultural cooperatives (ACs); drought risk management; smallholder farmers; livelihood assets; Cambodia

## 1. Introduction

Climate change already exists in Cambodia, where variations in rainfall patterns [1] manifest as flood and drought events [2,3]. In agriculture, rainfall changes have experienced three characteristics: a late onset, prolonged dry spells during the wet season, and early cessation. However, the country has produced no scientific studies on this characteristic [4,5], a fact that has been confirmed by others [6,7]. While floods have been annual events since 1999, drought events brought even more concern in 2003 and 2004 [8,9]. Both extreme events are the result of severe fluctuations in weather [10], and these climate-driven hazards have brought increased hardships and contributed to poverty traps [11], food insecurity [12], unsustainable livelihoods [13], migration [14], and repetitive business interruptions [15] of smallholder rice farmers [16,17]. The most recent study by Sok et al. (2021) [18] confirmed that between 1994 and 2018, drought events were more severe than flood events in terms of their adverse effect on rice production in the Mekong River and at Tonle Sap Lake. Similarly, Chann et al. [19] also observed severe and prolonged

droughts [19]; these events caused negative impacts on rice production in terms of damage and losses [20–25]. In 2004, the drought events decreased rice production by 82% [26], and in 2009, the Royal Government of Cambodia (RGoC) paid USD 12 million in order to recover from these significant events [27]. Less extensive drought events also adversely affected rice production in 1999, 2000, 2008, and 2012 [25]. In 2012, drought events triggered substantial damage, disturbing 7.5% of the rice cultivation area, and the impact increased to 11.5% in 2015. In 2016, 7.3% of the rice cultivation area was affected by drought events [28].

In Cambodia, the damage and losses incurred by the impacts of drought to the economy, society, and to the environment have recently been well-recorded by scholars and government agencies. Burkley et al. [29] found that the demise of Angkor was interspersed with drought events. In 1431 CE, the Khmer Empire's collapse was proven to be due to exposure to climate change and geopolitical/socio-economic pressures rather than rising conflict with the Siamese Kingdom (or Thailand) [30]. Evans et al. [31] suggested that an ineffective adaption to the prolonged droughts in mainland Southeast Asia during the transitional period from the Medieval Climate Anomaly to the Little Ice Age, in addition to social vulnerabilities, led to the collapse of Angkor. Drought was a prolonged and highly intensified hazard that stressed water supply and agricultural productivity. Droughts are among the deadliest threats when compared to other disasters such as floods and storms [30,32].

However, drought impacts have been addressed by the RGoC and by non-governmental organizations (NGOs); the role of community-based organizations (CBOs) is also emerging in terms of playing a role and taking a responsibility for empowering communities and prompting agricultural development. Since 2002, decentralized local institutions have contributed to the livelihood development of Cambodian people. Local institutions such as CoCs, NGOs, and CBOs are the providers of public and social services that are provided in response to local needs and the requirements for sustainable livelihoods [33], based on a bottom-up approach [34]. For example, the operation of community fishery in the Krakor district of Pursat Province helped fisherfolk to secure space for patrolling and saving groups [35]. In the Kampong Tralach district of Kampong Chhnang Province, community fishery has allowed indigenous Cham fisherfolk to use better resources, better sizes of fishing gear, and increased community outreach. Community-based eco-tourism (CBET) has also allowed local people to be involved in wildlife conservation and income generation by creating leisure activities. Local people have created services to attract tourists for income, such as camping with local guides, cycling, boat trips, hiking, and riding in ox carts [36]. In developing and low-income countries, ACs are established as CBOs in order to improve the livelihood of smallholder farmers [37]. This is achieved through access to agricultural services and investments [23], as well as through the diversification of crops [38]. Our research explores the experience and potential of ACs in the context of whether they can integrate climate change responses with specific cases of drought events into their operation.

AC operation is a long-term government investment with the aim of promoting agricultural development. It has been a promising method for advancing agriculture, as stated in the Agricultural Sector Master Plan 2030 (ASMP 2030). It is utilized to modernize widespread traditional subsistence agriculture practices [39]. Ofori et al. [40] witnessed that ACs in Cambodia diversified the agricultural horticulture of smallholder farmers and thus increased income. Since 1965, ACs have been established in different regimes under the Ministry of Agriculture, Forestry and Fisheries (MAFF) to promote local engagement [41] and to increase agricultural productivity and job alternatives [42] among smallholder farmers. In 2021, 1217 ACs were registered at the MAFF across the country, an increase from 1 AC in 2003, with 144,306 members comprising 53,703 males and 90,603 females [43]. In rural Cambodia, ACs are currently operating in order to support smallholder farmers to enhance technology transfer, knowledge sharing, and the exchange in crops and agricultural inputs, as well as access to loans and financial savings in rural communities. Moreover,

CBOs have worked to empower communities to manage and use natural resources for income generation activities and for environmental management [44,45].

This study has recognized the essential roles of CBOs, such as ACs, in empowering economic activities through agricultural productivity and building market channels. Moreover, this research produces evidence and scientific knowledge for the promotion of AC operations in Cambodia. For future implications, AC operations need to increase their roles and responsibilities in order to enhance the adaptive capacity of smallholder farmers to address the negative impact of climate change (especially in the case of drought events). Accordingly, this paper aims to analyze the functions of ACs in the practices of drought risk management among smallholder farmers by focusing on (1) the severity of drought events, (2) the effect of drought impacts on socio-economic development and livelihood, (3) the contribution of AC operations to drought management, and (4) the interactions between AC operations, adaptive capacity, and drought impacts. In the future, ACs may also play a role in building adaptive capacity among smallholder farmers and in establishing resilient communities through collective efforts to return to a reference state after drought disturbances.

## 2. Conceptual Framework

Scholars argue that the critical elements by which to mitigate the negative impact of drought events are found in a prompt response. This is achieved by implementing drought management strategies and action plans [46], as well as by conducting more proactive management in order to decrease emergency management costs [47]. While Deressa et al. [48] found that larger household sizes positively affect the adaptation to drought when the head of the household is male, Sadeghloo and Sojasi Qeidari [49] also added that older farmers should adopt drought management strategies. The IFRC [50] agrees that risk management at the local level is a crucial element for effective governance, the social fabric, and for building on the quality of community networks and disaster risk reductions, which the communities themselves undertake and which contribute to resilient and secure communities. This scenario indicates the need to determine the role of community participation in drought risk management [51]. Mazzucato and Niemeijer believe institutional support is significant for water resource management [52], adaptive capacity [53], and community development [54], specifically for improving sustainable livelihood through access to the five livelihood assets (human, nature, finance, the physical, and the social) [55]. It is thus argued that there is a need for the local government [56] and NGOs [57] to adopt increased roles in community development under a decentralized process, yet Tacconi [58] and Caplow [59] also argue that there is still too little attention paid to the social aspect of the five livelihood assets.

In recent decades, drought events have frequently affected extensive and vast areas of the world [60]. In the Asian Pacific region, drought impacts have been recorded as growing over the years due to climate variabilities, such as global warming, and due to the impact of El Niño and increasing environmental degradation [61]. In Asian and Sub-Saharan Africa, drought has become the major constraint to rice production in rain-fed areas [62]. The recent droughts in the Lower Mekong Basin (LMB) have severely affected crops and livelihoods. They have also contributed to lowered drinking water supplies and irrigation [63]. In China, the number of death tolls was reported to have declined temporally from 3 million people in 1928 to about 100 thousand in the 1980s [64]. However, the most recent and deadliest drought hit Somalia in 2011 and caused death tolls in the tens of thousands. Adamson and Bird [65] found that abnormally low flows in the mainstream of the Mekong River caused saline intrusion in the delta of Vietnam, thus bringing substantial economic losses globally. Additionally, they claimed that the low flows were implicated in the lower fish productivity in the inland fishery of Cambodia, which was especially the case in the great lake. The impacts of drought events that result in disaster are known as socio-economic and political drought events [66]; however, they can also be understood through vulnerability and social–ecological systems [53]. A drought is an event of prolonged shortages in the water

supply, and it is considered one of the most frequent and costliest natural hazards [60,67]. Drought management has been developed since the 1979s [68], which is when traditional drought management policies and practice were becoming increasingly challenging to achieve [69]. Drought is defined as water stress occurring mainly due to a lack of rain. In rain-fed ecosystems, drought is the greatest obstacle to rice production. For example, in the eastern Indian states of Jharkhand, Orissa, and Chhattisgarh, yield losses from severe drought (which occur in approximately one in every five years) averaged at 40% and were valued at USD 650 million [70]. Hooshmandi [71] reported a 27% mean yield reduction in rice due to these stress conditions.

The existing studies have investigated the operation of CBOs in aiding with drought impacts in the context of (1) adaptive capacity in Costa Rica [72], Cambodia [73], Kenya, and the Central Africa Republic [74]; local participation in Cambodia [41] and Indonesia [75]; and mitigation interventions in Rwanda [76], Kenya [77], Brazil [78], Indonesia [79], and Southern Ethiopia [80]. There has been a recognition of local capacity and the roles of CBOs in drought risk management [74,81,82]. Since ACs work with farmers, these CBOs are the key actors in disaster risk management, which includes managing for drought events. However, ACs mainly work to improve local livelihoods through raising awareness and bringing increased access to agricultural inputs and credits. For example, ACs are operating to strengthen agricultural development by providing services to support and serve smallholder farmers [83]. In China, ACs increase the yield of apple farmers when they participate in their activities [84], and they also contribute to the sustainable livelihood of rural communities in Zimbabwe through poverty re-education and improved food security [85]. ACs in India promote ecological resilience by increasing access to agricultural inputs and farmers' crop productivity [86]. To the authors' knowledge, and as per the current impact of climate change in Cambodia, ACs have essential tasks in response to the adverse effects of climate change because the agricultural sector is directly affected by weather extremes. Therefore, a hypothesis is drawn in this research: AC operation not only influences smallholder farmers' participation in their activities but also contributes to adaptive capacity and ameliorating drought impacts. In the rural communities of Benguet, the Philippines [87], ACs have worked to reduce disaster impacts among the affected households with credits and loans. In Cambodia, ACs are working with smallholder farmers to promote greater income generation. Therefore, they are important actors in terms of the response to disasters as their actions reduce risk and mitigate climate change impacts.

## 3. Materials and Methods

This research applied mixed methods in its study and involved collecting, analyzing, and interpreting quantitative and qualitative information and data [88]. Both quantitative and qualitative data were collected from the Bakan district in Pursat Province and the Barsedth district in Kampong Speu Province between February and May 2022. The quantitative research determined the perception, and it included the tests that were carried out to formulate the hypotheses [89] posted in the conceptual framework. We used Cochran's (1977) [90] formula to calculate the sample size for the unknown population, with the alpha level, a priori, at 0.05. We planned to use a proportional variable to set the acceptable error level at 5% and estimated the standard deviation of the scale to be 0.5. As an example of its use, Cochran's sample size formula is presented here in addition to an explanation of how these decisions were made:

$$n = \frac{Z^2(p*q)}{e^2} = \frac{(1.96)^2(0.5*0.5)}{(0.05)^2} = 384$$

where $Z$ represents the value for a selected alpha level of 0.025 in each tail, $Z = 1.96$ (the alpha level of 0.05 indicates the level of risk the researcher was willing to take; the true margin of error may exceed the acceptable margin of error), and $(p*q)$ represents the estimate of variance, $(p*q) = 0.25$.

For samples, this study recruited smallholder farmers from two districts (209 from the Bakan district and 212 from the Barsedth district). Bowerman et al. [91] suggested a sample size of 196 respondents for an unknown population, as is stated in the below formula. Based on the conditions above, a total of 421 smallholder farmers from the population of Cambodian agricultural contexts in Pursat and Kampong Speu Provinces are represented.

$$n = p(1-p)\left(\frac{Z_{\alpha/2}}{B}\right)^2 = 0.5(1-0.5)\left(\frac{1.96}{0.07}\right)^2 = 196$$

where $p$ is the probability, which is equal to 0.50; $Z_{\frac{\alpha}{2}}$ is the confidence interval of significance at 1.96; B is the tolerance error at 0.07 (7%).

McMillan and Schumacher [92] suggest using qualitative research as an inductive process by which to organize data into categories, as well as a way in which to explore the relationships among categories. For qualitative data, social and participatory tools (Appendix A) were applied, including the use of key informants at national levels, such as officers from the Ministry of Agriculture, Forestry and Fisheries (MAFF) and Heifer International Cambodia. At the district level, we also interviewed the District Office of Agriculture, Forestry, and Fisheries in the Barsedth district, as well as the district officers and those on commune councils (CoCs) in the Bakan district. To understand the contributions of AC operations, we conducted in-depth interviews with AC committees in the Bakan district (Chamreun Pal Agricultural Cooperative, Ponlue Agricultural Cooperative, and Preah Mlu Meanchey Agricultural Cooperative) and in the Barsedth district (Agri-Productive Transport vehicle: non-cold Chain Truck and Chamrostean Agricultural Cooperative). Moreover, a consultive meeting was held at the Bakan district office with 60 participants; these included district officers, CoCs, village heads, AC committees, and smallholder farmers. The meeting was organized to present the preliminary findings, collect feedback, discuss policy applications, and discuss planning. The presentation took the form of a forum to facilitate interaction between the district officers, CoCs, village heads, people, and the researchers regarding the research findings, as well as for the purposes of validation and clarification. Qualitative data and information were collected to describe the characteristics of droughts, drought impacts on socio-economic development and livelihood, and the contribution of AC operations to drought management. Qualitative data were also applied to explain the interactions between AC operations, adaptive capacity, and drought impacts.

We used a multiple regression model, a drought susceptibility model (DSM), and structural equation modeling (SEM) for the data analysis. The appropriate variables included in this survey were derived from discussions and from the agreements made between the key stakeholders, such as government agencies and non-governmental organizations (NGOs), for the purposes of running the multiple regression model, the DSM, and SEM. Multiple regression, which is a statistical technique, was used to predict the relationships between the services that are delivered by ACs to support smallholder farmers, as well as the nature of the access to natural assets, access to human assets, access to physical assets, access to social assets, access to financial assets, and access to water from January to May. We also used raw data regarding rainfall (from the Department of Meteorology in Cambodia) and supplementary irrigation data for between 2003 and 2020 from the Commune Database Online (CDB) of the Ministry of Planning (MoP). A drought susceptibility model was used for the drought susceptibility index (DSI), as is detailed in the following equation:

$$DSI = \frac{1 + \left(\frac{A_{area\ irrigated}}{A_{total\ area}}\right)}{1 + \left(\frac{Y_{commune}}{Y_{max}}\right)} - 1$$

where $A$ represents the rice cultivation area in the wet season and $Y$ represents the yield of rice production at the commune level.

SEM was applied to test the hypothesis that "AC operation not only influences small-holder farmers' participation in their activities, but also contributes to adaptive capacity and drought impacts" as follows:

A confirmatory factor analysis (CFA), or the measurement model, was used in this study. In line with the guidelines of Anderson and Gerbing [93], construct validity was applied for the evaluation. First, the exploratory factor analysis for all variables resulted in factor solutions, a result which was expected theoretically. Cronbach's alpha coefficients for each factor were greater than 0.60. Second, the CFA was used to assess the convergent validity of the measures. In this paper, there were two stages of the CFA: (1) the first-order factor model; and (2) the second-order factor model [94]. We adopted the first-order factor model to scrutinize the research constructs individually. Table 1 describes the threshold values of the CFA and SEM that were set to evaluate the CFA and SEM results (Table 2). The results of the first-order factor model reveal that all the threshold values were particularly satisfied (Figures A1–A4. The second-order factor model was then also selected to scrutinize the fitness of the overall model. When all loadings exceeded 0.60, each variable thus resulted in a t-value exceeding 1.96 ($p < 0.05$). Therefore, the CFA criteria were also satisfied. The overall goodness-of-fit assessment shows that $\chi^2/df = 1.461$, GFI = 0.897, AGFI = 0.872, NFI = 0.951, CFI = 0.984, and RMSEA = 0.033 (Table 1 and Figure 1). These results indicate that the model had a good model fit with satisfactory convergent validity. Since all values exceeded the established cutoff criteria, this analysis proceeded with a hypothesis test using SEM (Table 2).

**Table 1.** The thresholds of the CFA and SEM.

| Model Fit Statistics | Rule of Thumbs |
| --- | --- |
| $\chi^2/\text{D.F}$ | <3 |
| GFI | ≥0.90 |
| AGFI | ≥0.90 |
| NFI | ≥0.90 |
| CFI | ≥0.90 |
| RMSEA | <0.08 |

Note: $\chi^2$ = Chi-square; D.F = degree of freedom; GFI = goodness-of-fit; AGFI = adjusted goodness-of-fit; NFI = normalized fit index; CFI = comparative fit index; RMSEA = root means square error of approximation. Sources: Anderson and Gerbing [93], Jöreskog et al. [95]; Jöreskog and Sörbom [96,97]; Kline [98]; and Hooper et al. [99].

**Table 2.** The results of the CFA second-order factor model.

| Indicators | | Research Constructs | Standardized Loading > 0.60 | *t*-Value > 1.96 | AVE > 0.50 | CR > 0.70 |
| --- | --- | --- | --- | --- | --- | --- |
| ADS45_1 | ← | Adaptive capacity | 0.898 | A | 0.708 | 0.905 |
| ADS45_2 | ← | | 0.933 | 24.865 | | |
| ADS45_5 | ← | | 0.691 | 16.74 | | |
| ADS45_6 | ← | | 0.822 | 13.398 | | |
| IDL39_6 | ← | Drought impacts | 0.818 | A | 0.793 | 0.910 |
| IDL39_5 | ← | | 0.928 | 22.573 | | |
| IDL39_4 | ← | | 0.891 | 22.041 | | |
| IAC55_5 | ← | Participation in AC activities | 0.828 | A | 0.775 | 0.939 |
| IAC55_4 | ← | | 0.905 | 32.492 | | |
| IAC55_3 | ← | | 0.946 | 24.669 | | |

**Table 2.** *Cont.*

| Indicators | | Research Constructs | Standardized Loading > 0.60 | *t*-Value > 1.96 | AVE > 0.50 | CR > 0.70 |
|---|---|---|---|---|---|---|
| IAC55_2 | ← | | 0.879 | 22.415 | | |
| IAC57_4 | ← | AC operation | 0.867 | A | 0.728 | 0.960 |
| IAC57_3 | ← | | 0.868 | 28.363 | | |
| IAC57_2 | ← | | 0.867 | 24.397 | | |
| IAC57_1 | ← | | 0.84 | 23.012 | | |
| IAC57_5 | ← | | 0.929 | 28.041 | | |
| IAC57_6 | ← | | 0.912 | 27.156 | | |
| IAC57_7 | ← | | 0.826 | 22.345 | | |
| IAC57_8 | ← | | 0.844 | 22.956 | | |
| IAC57_9 | ← | | 0.707 | 17.361 | | |
| | | Goodness-of-Fit Index (Results) | | Goodness-of-Fit Index (Threshold) | | |
| | | $\chi^2$/D.F = 2.101 | | $\chi^2$/D.F < 3 | | |
| | | GFI = 0.932 | | GFI $\geq$ 0.90 | | |
| | | AGFI = 0.904 | | AGFI $\geq$ 0.90 | | |
| | | NFI = 0.963 | | NFI $\geq$ 0.95 | | |
| | | CFI = 0.980 | | CFI $\geq$ 0.95 | | |
| | | RMSEA = 0.051 | | RMSEA < 0.08 | | |

Note: A = parameter regression weight that was fixed at 1.000.

Furthermore, the average variance extracted (AVE) and composite reliability coefficients (CRs) were used to relate the quality of a measure. To avoid misconceptions, it was required to appropriately understand the equations of the AVE and CR, as well as to understand their association with the definition of validity and reliability. We explained, using simulated, one-factor models, how the number of items and the homogeneity of factor loadings influenced the AVE and CR results:

$$\text{AVE} = \frac{\sum_{i=1}^{n} \lambda_i^2}{n} \tag{1}$$

$$\text{CR} = \frac{\left(\sum_{i=1}^{n} \lambda_i\right)^2}{\left(\sum_{i=1}^{n} \lambda_i\right)^2 + \left(\sum_{i=1}^{n} \delta_i\right)} \tag{2}$$

where $\lambda$ (Lamda) signifies the standardized factor loading, $i$ is the number of items (1), and $\delta$ (Delta) refers to error variance terms (2), while $\delta = 1 - \lambda_i^2$.

According to Fornell and Larcker [100] and Peterson and Kim [101], the AVE must exceed 0.50, and the CR must exceed 0.70, respectively. Hair, Black, Babin, and Anderson [97] recommend that the *t*-value should be deemed significant when it is more than 1.96 and when the *p*-value < 0.05. All other criteria shown in Table 1 were also required to evaluate the results of the CFA and SEM. The AVE of the variables of adaptive capacity had an AVE = 0.708 and a CR of 0.905; the drought impacts had an AVE = 0.793 and a CR = 0.910; the participation in AC activities had an AVE = 0.775 and a CR = 0.939; and the AC operation had an AVE = 0.728 and a CR = 0.960, which met the thresholds and thus satisfied the overall model fit assessment as a good fit in terms of a GFI = 0.932, AGFI = 0.904, NFI = 0.963, and a CFI = 0.980, respectively.

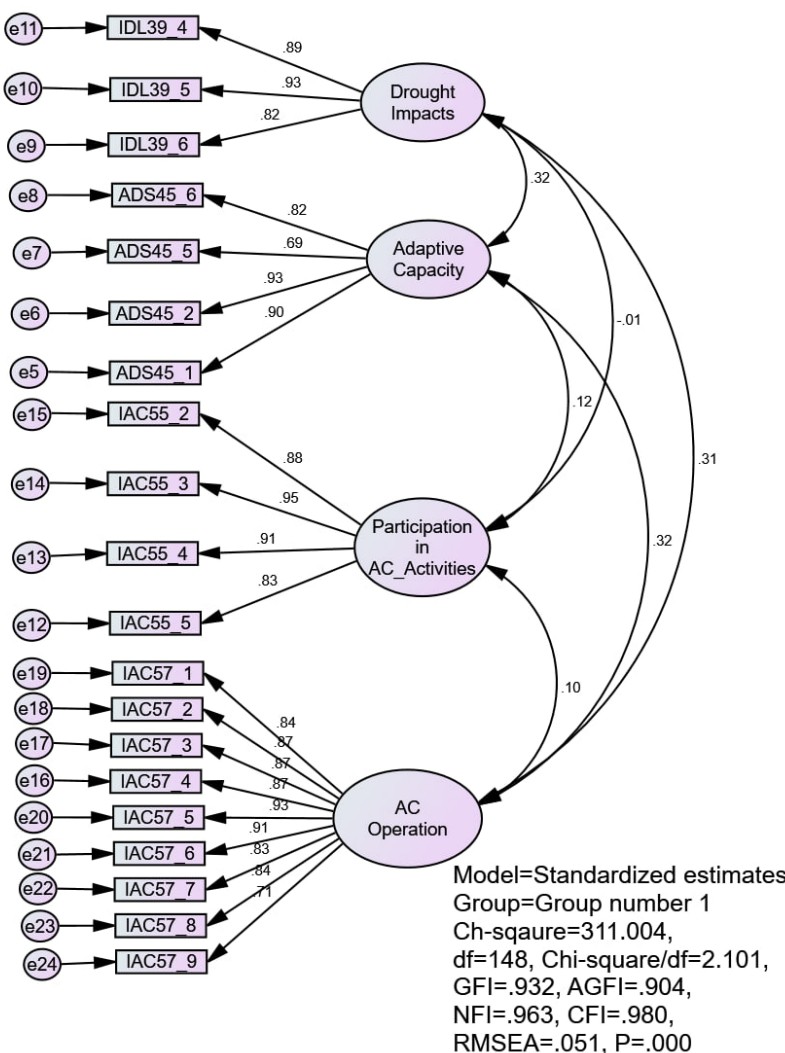

**Figure 1.** The results of the second-order factor model and overall model.

## 4. Results

### 4.1. The Severity of the Drought Events in the Kampong Speu and Pursat Provinces

Figure 2 compares the drought situation in Kampong Speu and Pursat Provinces regarding the average rainfall, irrigation ratios, and drought susceptibility indexes (DSIs). In this analysis, supplementary irrigation refers to wells, pumps, ponds, drip, and sprinklers; these were found to be widely accessible by smallholder farmers in the two provinces studied. In Cambodia, smallholder farmers consider drought impacts when rainfall distribution is interrupted during cultivation. Drought causes water shortages to affect the paddy fields. If smallholder farmers used six-month rice varieties, it would take an entire wet season, from seeding to harvesting between late May and early November. According to Chhinh and Millington [21], Cambodian smallholder farmers may face all stages of drought events, including early, middle-, and end-season drought events (Table 3). In other words, smallholder farmers were highly susceptible to drought events when they are without access to supplementary irrigation because access to medium-scale irrigation would remain limited; they saved their paddies when there was insufficient rainfall.

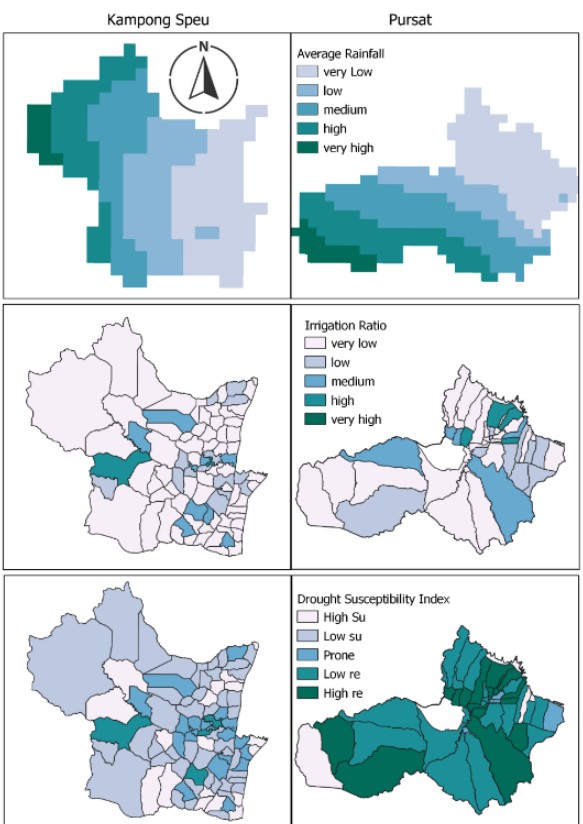

**Figure 2.** The drought context in Kampong Speu and Pursat Provinces.

**Table 3.** Generic rice cropping calendar and possible drought occurrence.

| Jan | Feb | Mar | Apr | May | Jun | Jul | Aug | Sept | Oct | Nov | Dec |
|---|---|---|---|---|---|---|---|---|---|---|---|
| | Dry Season | | | | Early Season | | Mid-Season | | End-Season | | Dry |
| Most of the paddy fields are not being used. | | | | The start of wet rice production. | | | The growth of the rice. | | The harvest period | | |

*4.2. Impact of the Drought Events on Socio-Economic Development and Livelihood*

The result of the household survey in the two districts, illustrated in Figure 3, shows how smallholder farmers assessed a moderate degree of the impacts of drought on their socio-economic development (WAI = 0.46). Smallholder farmers rated a high degree for the impact of drought when there was a reduction in household income (WAI = 0.64) and a moderate degree of impact for health (WAI = 0.60), loss of employment (WAI = 0.59), threatened household food security (WAI = 0.56), migration (WAI = 0.51), food scarcity (WAI = 0.50), limited food preference (WAI = 0.50), homelessness and sense of loss (WAI = 0.47), and a reduction in spending on festivals (WAI = 0.43). At the same time, smallholder farmers assessed a low degree of impact of drought on affected schooling for children (WAI = 0.37) and conflicts for water in society (WAI = 0.29). The results of the focus group discussions revealed that the smallholder farmers relied upon subsistent livelihoods in the two districts. When climate change places more pressure on agriculture, smallholders become more vulnerable, especially with respect to the impacts of drought. The farmers described to a higher degree that they depended on rain for their rice cultivation, wherein the higher degree that was advised meant that they were more vulnerable to climate-related hazards [Pers. Comm. FGD1 and FGD2].

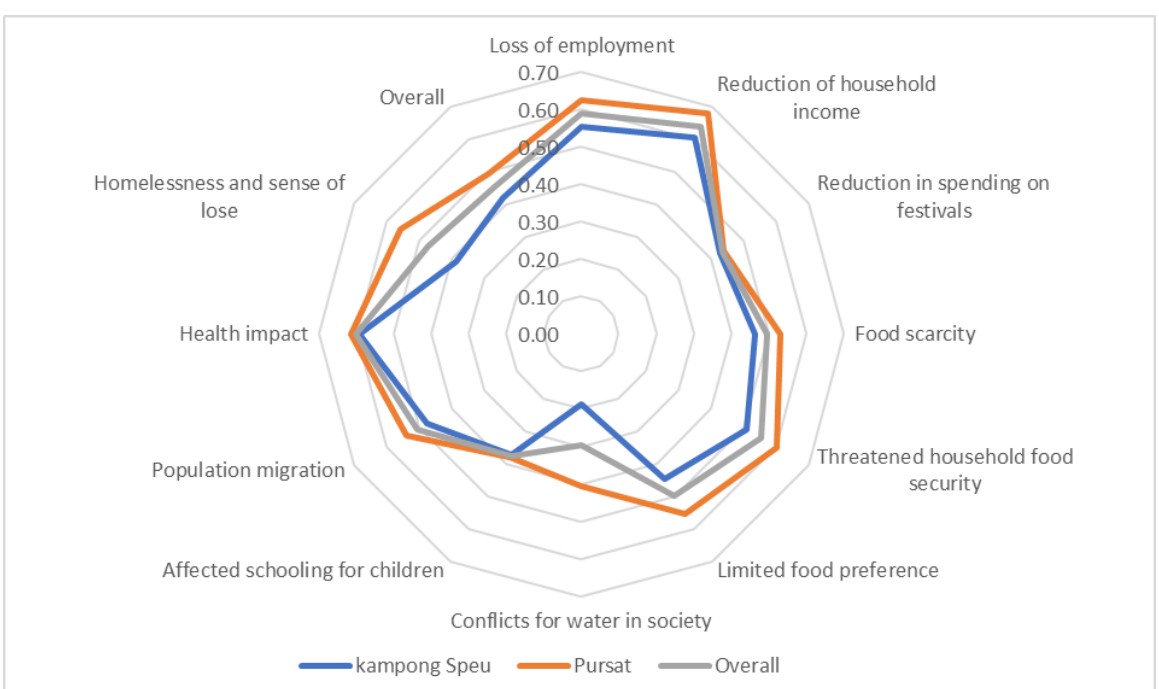

**Figure 3.** Impacts of drought on socio-economic development. Notes: WAI = the weight average index that is applied in order to be measured on a five-point scale (very low (VL) = 0.00–0.20, low (L) = 0.21–0.40, moderate (M) = 0.41–0.60, high (H) = 0.61–0.80, and very high (VH) = 0.81–1.00). OA = Overall assessment. *p*-value <0.05. *p*-value: The overall values for drought impacts = 0.000; loss of employment = 0.002; reduction in household income = 0.000; reduction in spending on festivals = 0.637; food scarcity = 0.005; threatened household food security = 0.000; limited food preference = 0.000; conflicts for water in society = 0.000; affected schooling for children = 0.735; population migration = 0.012; health impact = 0.346; and homelessness and sense of loss = 0.000.

The *t*-test analysis also revealed that the smallholder farmers in the Bakan district shared a higher degree of the impact of droughts on their loss of employment (*p*-value = 0.002), reduction in household income (*p*-value = 0.000), food scarcity (*p*-value = 0.005), threatened household food security (*p*-value = 0.000), limited food preference (*p*-value = 0.000), conflicts for water in society (*p*-value = 0.000), migration (*p*-value= 0.012), and homelessness and sense of loss (*p*-value = 0.000). In the Barsedth district, the smallholder farmers rated homelessness and sense of loss (WAI = 0.000) to a lower degree. Conflict for water became common in the Bakan district (WAI = 0.41), especially during the dry season; however, this was not a pressing issue in the Barsedth district (WAI = 0.18). Smallholder farmers in the two studied districts shared similar views regarding the reduction in spending on festivals (*p*-value = 0.637), their affected schooling for children (*p*-value = 0.735), and health impacts (WAI = 0.346).

During a consultative meeting, CoC members, AC committees, and smallholder farmers pronounced that there were water wars during the dry season. Water for paddy fields was not equally distributed among smallholder farmers in the district. Those residing near the main irrigation could pay to gain water from supplementary irrigation [Consultative Meeting]. The capacity to access water depended on their willingness to pay for gasoline and pump machines; individual smallholder farmers used their methods and investment to obtain water for their paddy fields [Pers. Comm. Interview-1]. A district officer in the Bakan district worked with smallholder farmers on these issues, but they could not solve the water conflicts. Each farmer wished to not share the water, and they did their best to obtain water for their own paddy fields as soon as possible due to their resources. The farmers in the upper parts of the district faced difficulty in receiving sufficient water because the lower part had already blocked or pumped water for their consumption [Pers.

Comm. K-3]. A commune council (CoC) in the Bakan district blamed the mismanagement of irrigation and lack of cooperation between the smallholder farmers as the cause of the water problem. Smallholder farmers did not cooperate with local authorities; they simply cared for their own paddy fields [Pers. Comm. K-5].

Comparatively, the droughts affected the socio-economic development of smallholder farmers in the Bakan district more than those in the Barsedth district ($p$-value = 0.000). The household survey shows that 77.6% of the respondents were employed as smallholder rice farmers, whereby 80.9% were in the Bakan district and 74.3% were in the Barsedth district. Their incomes primarily relied upon rice cultivation. When drought events occurred, it highly impacted their revenues. In the Bakan district, rice cultivation was crucial for people because most individuals owned agricultural lands. Moreover, people on the community wished to continue their traditional occupation [Pers. Comm. K-3]. A committee member of the Chamreun Pal Agricultural Cooperative described the impacts of drought in 2019, which destroyed rice fields, leading to an economic loss for smallholder farmers. Drought was becoming an increasingly severe hazard [Pers. Comm. Interview-2]. In the Barsedth district, many young people opted for rice farming as their chosen industry because there are around 200 factories in the Kampong Speu province. Thousands of jobs in non-farming industries are available for people in the Barsedth district. In addition, Barsedth is just 83 km from Phnom Penh, where there are good road conditions and transportation that facilitate people with a connection to the capital for this district. The alternative income of household members from non-farm sources reduced their vulnerabilities and risks when drought events eventually occurred [Pers. Comm. K-5].

Overall, smallholder farmers rated a high degree of impact from droughts (WAI = 0.66) on their livelihoods; indeed, both districts shared this view (the Bakan district (WAI = 0.65) and the Barsedth district (WAI = 0.67)) ($p$-value = 0.064). Smallholder farmers assessed that there was a high degree of pasture degradation (WAI = 0.76); increased water demand (WAI = 0.71); excessive groundwater pumping (WAI = 0.71); an increase in average temperature (WAI = 0.70); crop failures (WAI = 0.67); drier surroundings (WAI = 0.65); a greater deterioration of water quality (WAI = 0.65); declining groundwater levels (WAI = 0.64); damage to wildlife and habitats (WAI = 0.64); water scarcity in surface water bodies (WAI = 0.63); stunning (WAI = 0.63); an increase in food prices (WAI = 0.62); and forest degradation (WAI = 0.61). They rated a moderate degree of impact from droughts on the poor health of animals (WAI = 0.60); the loss of livestock (WAI = 0.59); famine (WAI = 0.58); and malnutrition (WAI = 0.57). The T-test analysis revealed that smallholder farmers in the Barsedth district rated a higher degree of the impact of droughts on forest degradation ($p$-value = 0.014); water scarcity in surface water bodies ($p$-value = 0.029); increases in food prices ($p$-value = 0.001); the poor health of animals ($p$-value = 0.010); malnutrition ($p$-value = 0.001); increased water demand ($p$-value = 0.040); and excessive groundwater pumping ($p$-value = 0.040) (Figure 4).

Field observation showed that the Bakan district retained greener and fuller wetlands, which are adjunct to the Tonle Sap Lake. The Barsedth district had less forest coverage and more water scarcity in its surface water bodies. Many people reduced their involvement in rice cultivation and became more involved in vegetable growing and raising livestock instead [Pers. Comm. K-5]. A committee at the Chamrostean Agricultural Cooperative observed that farmers opted to consume water from underground pumping in the Barsedth district because smallholder farmers could construct wells or pump at their houses or paddy fields [Pers. Comm. Interview-5]. For a long time, smallholder farmers benefited from the availability of natural resources, especially water from the wetlands and rains. Still, climate change has recently caused more frequent droughts and water shortages. The current water shortage and droughts have seriously affected smallholder farmers' livelihood development [Pers. Comm. K-1].

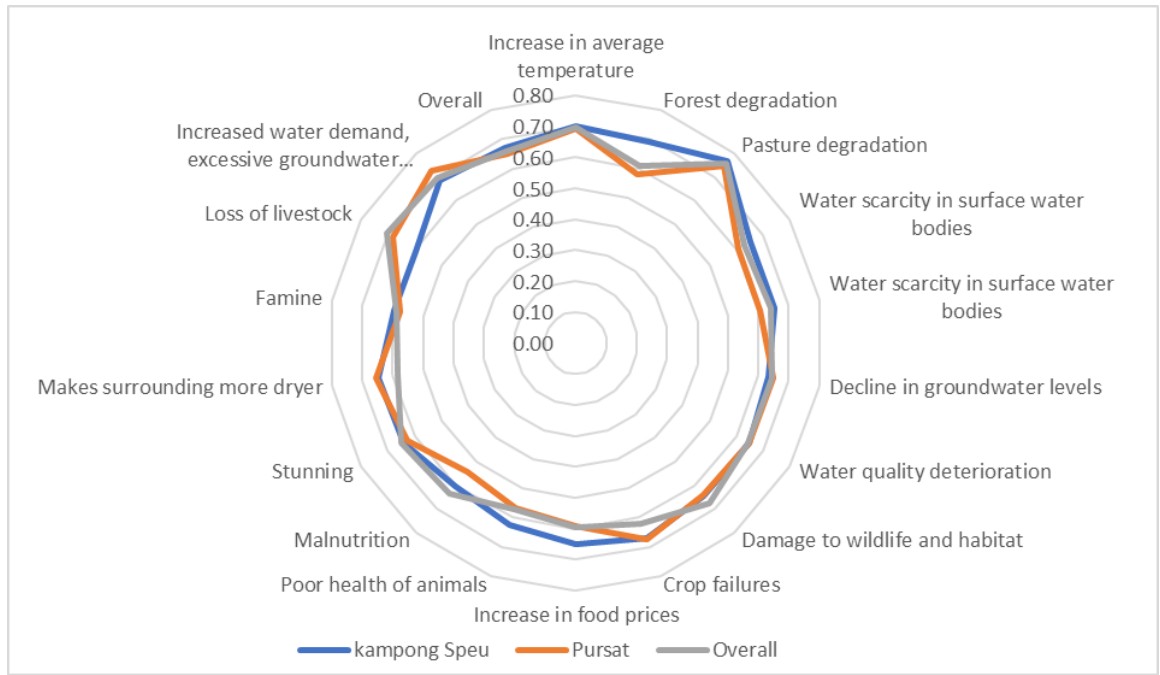

**Figure 4.** The degrees of drought impact on livelihood. Notes: WAI = the weight average index is applied to measure on a five-point scale (very low (VL) = 0.00–0.20, low (L) = 0.21–0.40, moderate (M) = 0.41–0.60, high (H) = 0.61–0.80, and very high (VH) = 0.81–1.00). OA = overall assessment. *p*-value <0.05. *p*-value: Increase in average temperature = 0.659; forest degradation = 0.014; pasture degradation = 0.363; water scarcity in surface water bodies = 0.029; the decline in groundwater levels = 0.392; water quality deterioration = 0.982; damage to wildlife and habitat = 0.961; crop failures = 0.931; increase in food prices = 0.001; poor health of animals = 0.010; malnutrition = 0.001; stunning = 0.560; making the surroundings more dry = 0.716; famine = 0.342; loss of livestock = 0.661; increased water demand; excessive groundwater pumping = 0.040; and the overall impact on livelihood = 0.064.

### 4.3. AC Support to Deal with Drought Risk Management

In the two districts studied, the group discussions found agreement that the increased access to the five livelihood assets helped smallholder farmers improve their adaptive capacity to reduce drought impacts [FGD 1 and FDI 2]. A committee at the Agri-Productive Transport vehicle: non-cold Chain Truck worked to improve access to natural assets, human assets, physical assets, social assets, financial assets, and water during the dry season because these factors were crucial for smallholder farmers' livelihood [Pers. Comm. Interview-4]. The ACs provided services for the smallholder farmers under the technical support of government agencies, CoCs, and NGOs. These institutions provided education on skill building and techniques and sought to bring the agricultural markets more in line with the private sector, such as supermarkets and wholesale sellers [Pers. Comm. K-2]. The multiple regression model shows that the services delivered by ACs helped to support smallholder farmers in accessing natural assets, with results of $\beta 1 = 0.226$ (22.6%), *t*-value = 4.366 > 1.96, and *p*-value =0.000 < 0.05. In addition, the physical assets returned values of $\beta 3 = 0.206$, *t*-value =4.100 >1.96, and *p*-value = 0.000 < 0.05. In contrast, the model predicted a significant negative contrition of the services that were delivered by ACs on social assets, with results of $\beta 4 = 0.156$ **, t-value =3.132 > 1.96, and *p*-value = 0.002 < 0.05. The services provided by ACs did not contribute to access to human assets, with results of $\beta 2 = 0.016$, *t*-value = 0.345 < 1.96, and *p*-value = 0.730 > 0.05. Moreover, the access to financial support delivered results of $\beta 5 = 0.072$, *t*-value = 1.545 < 1.96, and *p*-value = 0.123 > 0.05. Lastly, access to water from January to May for consumption delivered results of $\beta 6 = 0.077$, *t*-value =1.638 < 1.96, and *p*-value = 0.102 > 0.05 (Table 4).

Table 4. Services delivered by ACS to support smallholder farmers.

| Independent Variables | Dependent Variables | | | | | |
|---|---|---|---|---|---|---|
| | Services Delivered by ACs to Support Smallholder Farmers | | | | | |
| | Model 1 ($\beta_1$) | Model 2 ($\beta_2$) | Model 3 ($\beta_3$) | Model 4 ($\beta_4$) | Model 5 ($\beta_5$) | Model 6 ($\beta_6$) |
| Access to natural assets | 0.226 *** | | | | | |
| Access to human assets | | 0.016 | | | | |
| Access to physical assets | | | 0.206 *** | | | |
| Access to social assets | | | | −0.156 ** | | |
| Access to financial assets | | | | | 0.072 | |
| Accessible to water from January to May | | | | | | 0.077 |
| $R^2$ | 0.058 | 0.058 | 0.099 | 0.125 | 0.130 | 0.136 |
| Adjusted-$R^2$ | 0.056 | 0.054 | 0.092 | 0.116 | 0.120 | 0.123 |
| F-value (Sig. $p$-value < 0.05) | 25.655 | 12.866 | 15.059 | 14.653 | 12.324 | 10.760 |
| $t$-value > \|1.96\| | 4.366 | 0.345 | 4.100 | 3.132 | 1.545 | 1.638 |
| Sig. $p$-value <0.05 | 0.000 | 0.730 | 0.000 | 0.002 | 0.123 | 0.102 |

Note: *** $p < 0.001$ and ** $p < 0.05$. Hair et al. [97] suggested that when the t-value is more significant than 1.96 and when the significance of the $p$-value is less than 0.05, the proposed research hypotheses can be accepted. In contrast, the proposed research hypotheses would be rejected if the $t$-value <1.96 and the $p$-value > 0.05.

The model revealed that the services delivered by ACs positively contribute to the access to natural assets (i.e., lakes, swamps, wells, ponds, streams, flood forests, and open access resources) and physical assets (i.e., roads, bridges, river ports, irrigations, local markets, health facilities, and school facilities for children). In the Bakan district, the ACs raised awareness and capacity building among the smallholder farmers with respect to the importance of natural assets (i.e., lakes, swamps, wells, ponds, streams, flood forests, and open-access resources). Furthermore, the knowledge and skills were helpful for smallholder farmers to become more involved in sustainable resource management [Pers. Comm. Interview-3]. The ACs in the Barsedth district had a good relationship with CoCs and NGOs in terms of mobilizing the resources to improve physical assets [Pers. Comm. Interview-5]. In 2021, Heifer International Cambodia provided the Agri-Productive Transport vehicle: a non-cold Chain Truck with a truck. The truck helped to increase the AC's business activities for transporting the agricultural inputs of AC and for the farmers and buyers [Pers. Comm. Interview-4]. In 2021, the Accelerating Inclusive Markets for Smallholders (AIMS) also constructed a collection and distribution center of agricultural products to create a market chain; this allowed all farmers to distribute and sell their agricultural products [Pers. Comm. Interview K-5]. In the Bakan district, the ACs advised the CoCs and the District Office of Agriculture, Forestry, and Fisheries to construct irrigation and roads to support farmers in their access to water for their paddy fields, as well as to help better facilitate the transportation of agricultural products to local markets [Pers. Comm. K-3].

In contrast, the model predicted the negative contribution of ACs to social assets, such as raising concerns about water shortage, participating in the activities of NGOs, participating in the activities of CoCs, participating in the activities of government agencies, involvement in the activities of community fishery, participation in the activities of community forestry, and being involved in community decision making. The ACs have applied a participatory approach with support from NGOs; however, smallholder farmers still felt that their participation did not contribute much to the decision-making process for community development. An AC committee in the Chamrostean Agricultural Cooperative advised that smallholder farmers in the Barsedth district had more opportunities to

participate in the workshop and community meetings that were organized by government agencies and NGOs. Their participation contributed to the planning and policy implication, and they were able to raise their concerns and issues [Pers. Comm. Interview-5].

The local authority, agricultural officers, and smallholder farmers in the Bakan district agreed that the planning process at the CoC level was essential for integrating local needs and that smallholder farmers should be invited to participate. However, the local needs raised during the CoC meeting could not be fulfilled because the annual budget for the commune investment plan was mainly used for physical infrastructure only. In most cases, the yearly commune investment budget did not allocate funds for supporting AC operations, social issues, or drought management [Pers. Comm. Interview-3]. Local authorities, agricultural officers, and smallholder farmers in the Bakan district agreed that ACs did not address the promotion of social trust very well, nor did they help with promoting a culture of sharing and helping each other. In Cambodia, farmers traditionally work together, which is consistent with a vital principle of the ACs [Consultive Meeting]. All the involved agencies mainly discussed and worked to increase access to irrigation, roads, microfinance, and natural resource management, but the scope of addressing social issues remained limited [Pers. Comm. FGD 2]. During a discussion in the Bakan district, the participants announced that there were water wars occurring between the smallholder farmers; in fact, these conflicts were not predominantly caused by a lack of irrigation but were instead due to the fact that they did not share the resource. Everyone felt as if they did not receive sufficient water; however, they did not work together to solve the problem [Pers. Comm. FGD 1].

Unfortunately, the ACs did not contribute to improving access to financial assets (i.e., access to microfinances for loans, access to a commercial bank for a loan, access to local lenders for loans, participation in saving groups, and access to income generation activities), nor to access to water resources for consumption during dry seasons, such as for bathing, drinking, cooking, washing, dry rice cultivation, and crop cultivation. All the ACs applied the same core principle of share sales between the members. The smallholder farmers could invest USD 24.5 per share as shareholders and could thus expect to receive annual profits. However, the smallholder farmers who bought shares from Agri-Productive Transport vehicle: a non-cold Chain Truck in the Barsedth district had not yet received benefits from their shares in 2022 due to the impacts of the COVID-19 pandemic. The focus of the discussion in the two districts was that the financial assets accessed by smallholder farmers through ACs remained small. The revenue produced by ACs through share sales and the distribution of agricultural inputs was not sufficient. Simultaneously, the annual benefits from shares and the available loans or savings from ACs were also small and thus made it harder to encourage people to participate. The contributions of smallholder farmers to other smallholder farmers were found to not be of much benefit. Instead, they preferred taking loans from microfinance or commercial banks with high interests because they could borrow with their demands for agricultural investment [Pers. Comm. FGD 1 and FGD 2].

Access to water was the most critical resource for smallholder farmers, especially during the dry season. In the consultative meeting, the available and accessible water for farming activities was discussed. Almost all the farmers depended on rain-fed agriculture. During the dry season, smallholder farmers had difficulty accessing water for their paddy fields. When drought events happened, as in 2018, for example, smallholder farmers needed to buy or rent pumps in order to obtain water from the nearby dikes, ponds, and cannels [Pers. Comm. Interview-1]. An officer at the District Office of Agriculture, Fishery, and Forestry in Barsedth district explained that the dikes, ponds, and cannels dried out faster and remained that way for several weeks or months. The rice of smallholder farmers who stayed away from water sources such as wetlands or cannels did not have sufficient water [Pers. Comm. K-5]. The ACs have worked closely with key stakeholders from government agencies and NGOs from district to commune levels on this issue; however, as of yet, access to water during the dry season remains unsolved [Pers. Comm. Interview-2]. In general,

access to water during the dry season was not stable and was uncertain; smallholder farmers were thus challenged to maintain productivity with a lack of water during the dry season. However, Cambodia's water from the Mekong River, Tonle Sap Lake, and other wetlands was still widely available. Still, making physical infrastructure public for smallholder farmers for the entire year has proven to be costly, especially in the dry season [Pers. Comm. K-1].

*4.4. Interactions between the AC Operations, Adaptive Capacity, and Drought Impacts*

After running the CFA, the same variables illustrated in Table 5 were also used to conduct an SEM analysis. SEM was used to test a hypothesis with the likelihood estimation method, and the SEM analysis supported the variables well. According to Anderson and Gerbing [93], the second-order factor model was adopted to test the overall variables. The results revealed that the goodness-of-fit measurements were satisfactory (GFI = 0.932, AGF = 0.904, NFI = 0.963, CFI = 0.980, and RMSEA = 0.051) (Table 2 and Figure 1). Furthermore, with the goodness-of-fit assessment, they also indicated that the model was acceptable [97]. The CFA, which applied the same variables shown in Table 2, was also conducted before proceeding with the SEM to test the likelihood estimation method. Table 5 and Figure 5 show that the goodness-of-fit measurements were satisfactory (GFI = 0.932, AGFI = 0.904, NFI = 0.963, CFI = 0.980, and RMSEA = 0.051), indicating that the model is acceptable in the face of a goodness-of-fit assessment. Overall, the adaptive capacity of smallholder farmers played a vital role in reducing drought impacts and enhancing AC operations. The SEM suggested that adaptive capacity had a positive and significant impact on drought impacts, as it delivered values of $\beta = 0.254$ ***, $p = 0.000 < 0.001$, and $t$-value = $4.511 > 1.96$. The participation in AC activities resulted in $\beta = 0.106$ **, $p = 0.04 < 0.05$, and $t$-value = $2.053 > 1.96$. At the same time, the AC operation significantly impacted the adaptive capacity, resulting in values of $\beta = 0.352$ ***, $p = 0.000$ ($p < 0.001$), and $t$-value = $6.957$ ($t$-value > 1.96). These factors also contributed to alleviating drought impacts, as can be seen in the values of $\beta = 0.196$ ***, $p = 0.000 < 0.001$, and $t$-value = $3.659$. However, the participation in AC activities had no significant effects on AC operations, which resulted in values of $\beta = -0.077$, $p = 0.122 > 0.05$, and $t$-value = $-1.546 < 1.96$. The results on the impacts of drought were $\beta = 0.051$, $p = 0.298 > 0.05$, and $t$-value = $1.1041 < 1.96$.

**Table 5.** The results of SEM.

| Constructs | | Indicators | Standardized Coefficient (β) | *t*-Value | *p*-Value |
|---|---|---|---|---|---|
| Adaptive capacity | → | ADS45_1 | 0.902 | A | *** |
| | → | ADS45_2 | 0.929 | 25.318 | *** |
| | → | ADS45_5 | 0.694 | 16.851 | *** |
| | → | ADS45_6 | 0.769 | 17.99 | *** |
| Participation in AC activities | → | IAC55_5 | 0.827 | A | *** |
| | → | IAC55_4 | 0.905 | 32.434 | *** |
| | → | IAC55_3 | 0.945 | 24.631 | *** |
| | → | IAC55_2 | 0.879 | 22.396 | *** |
| AC operation | → | IAC57_4 | 0.867 | A | *** |
| | → | IAC57_3 | 0.854 | 28.414 | *** |
| | → | IAC57_2 | 0.866 | 24.53 | *** |
| | → | IAC57_1 | 0.843 | 23.077 | *** |

**Table 5.** *Cont.*

| Constructs | | Indicators | Standardized Coefficient (β) | *t*-Value | *p*-Value |
|---|---|---|---|---|---|
| | → | IAC57_5 | 0.925 | 27.748 | *** |
| | → | IAC57_6 | 0.913 | 27.158 | *** |
| | → | IAC57_7 | 0.83 | 22.488 | *** |
| | → | IAC57_8 | 0.845 | 22.99 | *** |
| | → | IAC57_9 | 0.709 | 17.433 | *** |
| Drought impacts | → | IDL39_6 | 0.818 | A | *** |
| | → | IDL39_5 | 0.865 | 21.827 | *** |
| | → | IDL39_4 | 0.895 | 22.12 | *** |
| Path Relationships | | | | | |
| H1: Adaptive capacity → AC operation | | | 0.352 | 6.957 | *** |
| H2: Adaptive capacity → Participation in AC activities | | | 0.106 | 2.053 | 0.04 |
| H3: Adaptive capacity → Drought impacts | | | 0.254 | 4.511 | *** |
| H4: AC operations → Drought impacts | | | 0.196 | 3.659 | *** |
| H5: Participation in AC activities → Drought impacts | | | −0.077 | −1.546 | 0.122 |
| H6: Participation in AC activities → AC operations | | | 0.051 | 1.041 | 0.298 |
| Goodness-of-fit index | | | | | |
| $\chi^2$/D.F = 2.105 | | | | | |
| GFI = 0.932 | | | | | |
| AGFI = 0.904 | | | | | |
| NFI = 0.963 | | | | | |
| CFI = 0.980 | | | | | |
| RMSEA = 0.051 | | | | | |

Note: A = parameter regression weight was fixed at 1.000 with a significant *p*-value < 0.05 and a t-value > 1.96. *** *p* < 0.001.

In the rural communities of Cambodia, ACs were established to empower smallholder farmers to form groups to promote agricultural development and income generation activities. According to Heifer International Cambodia, NGOs have worked with government agencies and ACs to build capacity, infrastructure development, and marketing for agricultural products [Pers. Comm. K-2]. However, promoting agricultural development cannot be separated from disaster risk management, climate change adaptation, and resilience; in addition, flood and drought are the primary hazards affecting the livelihoods of smallholder farmers [Pers. Comm. K-1]. The participants in the consultative meeting agreed that AC operations did not directly impact drought risk management, but they did contribute to increasing the adaptive capacity of smallholder farmers. For example, if smallholder farmers could maintain a high crop productivity and have good markets for their agricultural products, they would be able to invest more in their paddy fields [Pers. Comm. K-3]. The agricultural officer agreed that wealthy farmers had a better and stronger adaptive capacity, and they were less vulnerable to drought. Wealthier smallholder farmers could respond to drought impact in a timelier fashion; for example, they could pay for gasoline and pump machines to fill water into their paddy fields. Some wealthier farmers could also afford to pay for drip or sprinkler irrigation and a net to protect their crops and vegetables from water shortages and insects [Pers. Comm. K-1].

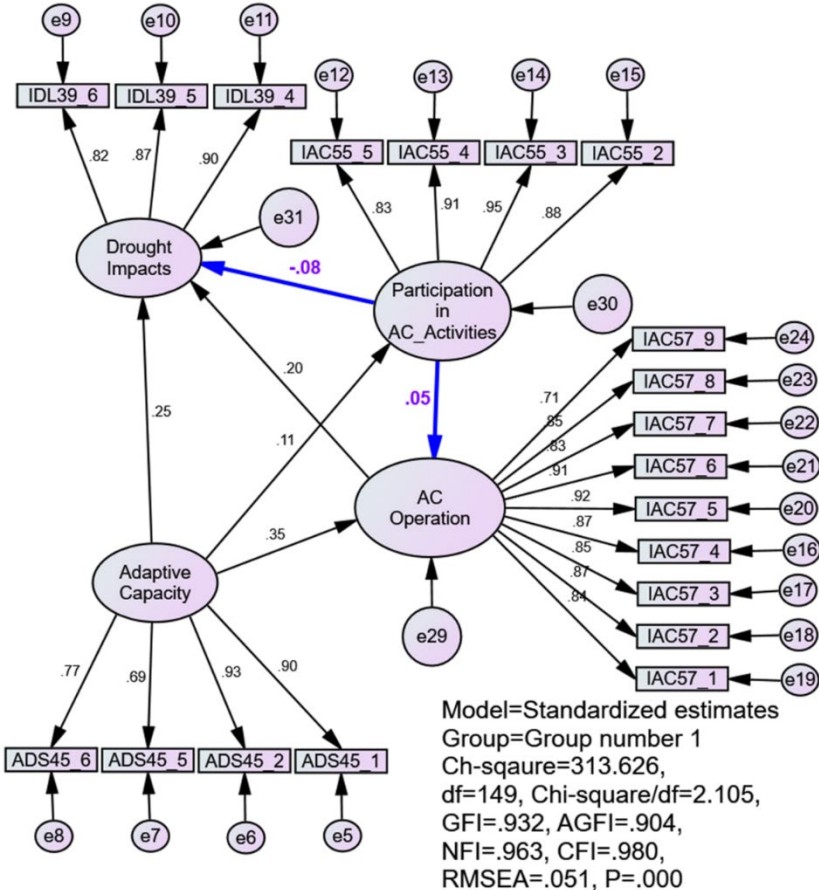

**Figure 5.** The results of the structural equation modeling (SEM).

All the ACs were cooperating with government agencies and NGOs to raise awareness and to provide skills and techniques to reduce the impacts of drought to some extent. Smallholder farmers were mainly involved in raising awareness, skill development, marketing, and income generation activities through AC operation, and they were also beneficial in terms of reducing the impacts of drought [Pers. Comm. FGD 2]. The officer in the Bakan district administration described the limitation of ACs as due to the fact that they did not have sufficient financial and human resources to support smallholder farmers in order to reduce drought impacts [Pers. Comm. K-3]. The community meeting discussed the relationship between participation in AC activities, adaptive capacity, and drought impacts. The participants recognized the roles of ACs in increasing adaptive capacity and reducing drought impacts. They compared the smallholder farmers who participated in ACs and those who did not; however, they had different degrees of awareness and responses to drought impacts. They argued that smallholder farmers who were engaged in ACs could better solve the problems faced during droughts than those who were not engaged in ACs. Through field observations in the Bakan and Barsedth districts, the smallholder farmers who participated in ACs obtained knowledge and techniques from agricultural offers. At their house ground, the NGOs have also started experimenting with growing vegetables to be resilient to droughts and shortages of water. Smallholder farmers have gathered banana leaves around their vegetables in order to wet the soil such that the crops did not require much water to grow.

## 5. Discussion

### 5.1. Human-Made and Climatic Factors Causing Drought Events

Comparatively, the Bakan district received an average rainfall of 1658 mm, and the Barsedth district received 1395 mm. Both districts are located in lower land areas, but

the smallholder farmers in the Bakan district (17%) had more access to supplementary irrigation than those in the Barsedth district (13%). As a result, the DSI analysis suggests that the Bakan district was less susceptible to drought than the Barsedth district. The analysis shows a strong relationship between the irrigation ratio of the commune and a higher drought resilience index at a *p*-value of 0.00 and an R-square of 0.7. The Barsedth district is more prone to drought than the Bakan district. Moreover, the Barsedth district has a lower yield of paddy; it is low by about 2.3 tonnes perhectare, while the Bakan district had 2.8 tonnes perhectare (with *p*-value = 0.00, using the *t*-test). Figure 6 suggests that only a few communes in Kampong Speu Province are resilient enough to drought, and these are mainly located in low-lying areas where they can access the supplementary irrigation system.

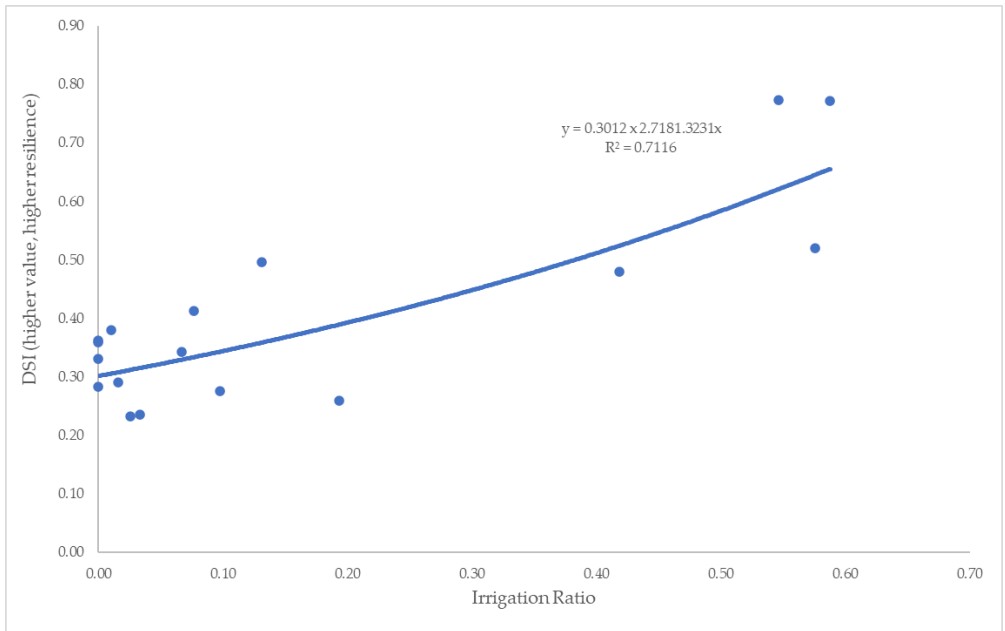

**Figure 6.** The relationship between irrigation ration by commune and their DSI value.

Raw data, surveys, and case studies confirm that drought impacts in Cambodia are associated with human-made and climate factors. The raw data regarding rainfall and supplementary irrigation prove that the Bakan district is more resilient to drought than the Barsedth district. However, the findings of the survey and case studies were found to contradict. The socio-economic development of smallholder farmers in the Bakan district was shown to have suffered more from drought impacts than development in the Barsedth district. The water shortages or droughts were primarily caused by water demands in which the rain and irrigation systems were fed to the water suppliers. In the Bakan district, rice cultivation during the dry season was approximately 12,200 hectares in 2020. In early 2013, the Bakan district had about 8000 hectares of dry rice. In contrast, in the same year, the Barsedth district saw significantly less dry rice cultivation of approximately 200 hectares. One angle of the conflicts was related to the cultivation of dry rice exceeding the irrigation capacity as farmers kept expanding the cultivation area. Other angles were the manner of water management and the social sodality and extent of cooperation. It was found that strong management for the irrigation scheme was required. From the focus group discussion, each time there was a conflict related to water sharing, the provincial government was required to solve the issue case by case.

The authors argue that the importance of supplementary irrigation is essential for resilience to drought impacts. Access to supplementary irrigation has improved smallholder farmers' resilience to drought, especially for the production of wet season rice. The supplementary irrigation could mitigate the effects of irregular rainfall during the early,

middle-, or late-season drought. Culturally, farmers in Cambodia begin rice cultivation during the early wet season. In the case of drought in late June or July, irrigation will supplement the water to the paddy fields. Cambodia's paddy field is always filled with rainfall during September and October. Thus, as long as smallholder farmers have access to supplementary irrigation during June and July, they can safely grow rice during the early wet season. Supplementary irrigation also plays a significant role in supplying water during the late wet season. However, the early cessation of rainfall causes disaster in rice production. With supplementary irrigation, smallholder farmers can use the water to secure that their rice is fully grown until the time of harvesting. In the context of rice production in Cambodia, the lack of water for supplementary irrigation is the primary source of drought susceptibility, which usually leads to less yield and is more prone to drought damage.

Both physical and financial assets are significant for promoting the economic activities of ACs and smallholder farmers. On the other hand, the economic empowerment of smallholder farmers is automatically beneficial in terms of reducing the impacts of drought. ACs with financial and technical support from local government (including district offices and CoCs) and NGOs can perform better in delivering their daily services to increase access to the human, natural, social, physical, and financial assets between smallholder farmers. During group discussions in the two districts, smallholder farmers identified the vital infrastructure required for agricultural development; these factors included irrigation, roads, and means of transportation. Smallholder farmers can connect the market channels and reduce transportation costs when the physical assets are increased. Physical assets increase the profits of smallholder farmers to earn and create bargaining powers with middlemen and with buyers. In Cambodia, district offices and CoCs worked to improve access to small-scale infrastructure developments with annual investment funds from the Ministry of Interior (MoI) under the Decentralization and Deconcentration (D&D) initiative. In addition, the NGOs constructed supplementary irrigation, especially dikes, ponds, wells, and pumps, in order to increase the access to water for smallholder farmers. Concerning financial assets, the primary sources for agricultural investments were mainly derived from microfinance, commercial banks, and local lenders, which are widely available across the country. The ACs could only aid with small loan savings through saving groups and shareholders, and only if more than USD 1000 were derived from microfinance, commercial banks, and local lenders with high interest rates.

*5.2. Why Is AC Operation Constraining Increasing Access to Livelihood Assets and Addressing Drought Impacts among Smallholder Farmers?*

The results of the quantitative and qualitative data analysis show that ACs, with support from local government (including district offices and CoCs) and NGOs, have been assisting smallholder farmers in increasing access to natural, human, and social assets through raising awareness (through workshops and community meetings) and capacity building (through skill training, exchange field visits, and coaching). Awareness-raising activities help increase social assets through local participation in the planning and decision-making processes and through promoting capacity and skill building; they are essential to improving human investments, especially in horticulture and livestock raising. At the same time, the participatory process and the awareness of smallholder farmers are crucial for empowering smallholder farmers to participate in mitigating the impacts of drought. In short, ACs have played more roles in facilitating access to the five assets of livelihood in the reduction of drought impacts than those found in implementing them alone. ACs worked and played roles on the behalf of NGOs and the Provincial Office of Agriculture, Fisheries, and Forestry in communities. Still, they are mostly unable to perform over long-term periods if there is no government and NGO support. The agricultural officer mentioned that almost all ACs were established with the support of financial and technical assistance from NGOs and that the MAFF was also providing them with legal backing, such as the registration process with which CBOs operate.

NGOs primarily support AC operations for three to five years. However, most ACs cannot continue after the project is completed. There are several reasons for their inability to continue their activities and services without external support; in the consultative meeting in the Bakan district, it was agreed that the human and financial sustainability of carrying out core activities and services was the core constraint. The focus group discussions in the two districts confirmed that the annual revenues and business activities of ACs are still small. In addition, they could not sufficiently allow ACs to carry out investment by themselves without external support. In 2020, smallholder farmers received USD 4.75 per share as their annual benefit [Pers. Comm. Interview-4]. The AC at the Chamrostean Agricultural Cooperative in the Barsedth district sold fertilizers and other agricultural inputs. For example, one fertilizer pack was sold for USD 45, and the AC earned USD 1.25 per pack. These ACs also created saving groups in which smallholder farmers could deposit money, land money, and share experiences about income generation activities [Pers. Comm. Interview-5].

The extent of willingness, commitment, and trust between the AC committee members was also found to be a constraint in terms of moving operations forward. The education level of most AC committee members was lower than the national 9-year education; they could not carry out activities without the support of the government and NGOs. Moreover, AC committee members did not have the initiative to innovate business activities and development services in order to support smallholder farmers in terms of sustainable livelihood from agricultural development and reduction in drought impacts. On the other hand, the average age of the members in the AC committees and the smallholder farmers was 50 years; it was also found that they did not have a high passion or willingness to expand their business, nor to develop their technology and skills. Both the AC committee and the smallholder farmers continued their agricultural activities as per their traditional and subsistent livelihoods. They were not very competitive in terms of productivity improvement, market expansion, and capital investment [Pers. Comm. Interview-5]. Trust building between the AC members and smallholder farmers remained a concern. There were experiences of committee members asking for AC money and shares and then leaving the communities. For example, The Agri-Productive Transport vehicle: a non-cold Chain Truck in the Barsedth district, which was established in April 2017, dissolved at the end of 2019 due to bankruptcy and due to the AC committee head leaving the community. There were 97 members, with 100 as shareholders at approximately USD 24.5 per share. In January 2020, the ACs restructured and restarted their operation with support from certain NGOs (such as Heifer International Cambodia). This NGO provided the AC with the capacity for building leadership management, skill building, and the facilities to start up businesses. A new structure with NGO support helped to renew trust between the smallholder farmers and to increase memberships and shares. By December 2022, this AC had a member budget of USD 1080 and was composed of 864 shareholders (including 558 women and 277 men), or USD 10,800 from 710 households.

*5.3. The Roles of ACs in Increasing Adaptive Capacity and Reducing the Impacts of Drought on Smallholder Farmers*

The survey found that drought greatly impacted the livelihoods of smallholder farmers and that AC operations do not sufficiently contribute to increasing the access to livelihood assets, which include human, financial, and social assets. In Cambodia, the five livelihood assets are essential to promoting the adaptive capacity of the communities of the Mekong River. With appropriate strategies, as well as with access to human, physical, and social assets, there would be an advancement made in the increase in financial and natural assets [102]. In the future, ACs in Cambodia will be required to improve access to the human and social assets of smallholder farmers; ACs should have their own technical and financial capacity to provide skill building and should organize social events to engage smallholder farmers in planning and policy implementations. If ACs continue to depend on financial and technical support from NGOs and local government, they cannot maintain

skill building for their members over more long-term periods. Therefore, ACs should be secured with financial and human resources in order to enable self-operation, especially for skill building, coaching, and the core activities required to carry out daily operations. ACs should improve their financial and technical capacities during NGO project implementation before the project is phased out. If possible, ACs should recruit young university students to support their daily operations, as their degrees helps with assisting the AC committees; this would include help with technology, training activities, monitoring, evaluation, and business analysis. The Memorandum of Understanding (MOU) between the ACs and the District Office of Agriculture, Fisheries, and Forestry could help create long-term cooperation for training, coaching, and technical support at the community level.

Adaptive capacity is a strategy based on autonomous ability, or it can be planned with timely preventative or reactive [103] measures for coping with harm or risk [104]. According to Watson et al. [105], adaptive capacity requires information exchange, technological advances, institutional arrangements, and the availability of finance to be suitable enough for integration into disaster preparedness [106,107]. To promote the adaptive capacity for drought in developing countries such as Cambodia, ACs play essential roles in planning and delivering services between smallholder farmers in order to increase human, natural, and social assets. The improved increase in the three previously mentioned livelihood assets by the ACs is found to empower smallholder farmers to be strong in their economic and social responsibilities through a participatory process. Sometimes, ACs are required to work with local governments, NGOs, and the private sector in order to increase access to physical and financial assets. Moreover, the annual revenues of ACs should also be allocated for development projects, especially for disaster risk management and other related social issues. In addition, ACs should advocate for CoCs and district offices to include the ACs' action in their annual investment plans, with specific budgets to promote disaster risk management, capacity building, skill and skill building.

## 6. Conclusions, Limitations, Further Research, and Implications

The findings in the Bakan district of Pursat Province and the Barsedth district of Kampong Speu Province, with insights into the impacts of AC operations in Cambodia, conclude the following: (1) The Bakan district was more resilient than the Barsedth district to drought in terms of higher annual average rainfall and in terms of the ratio of supplementary irrigation. The livelihoods of smallholder farmers in the Bakan district were affected by the high demands and water conflicts around paddy rice cultivation. Drought impacts in Cambodia are influenced by both climatic factors and human-made factors. This study recognizes the critical contribution of supplementary irrigation for improving the resilience of smallholder farmers to drought, especially for the purposes of dry season cultivation. (2) AC operations assisted smallholder farmers in accessing natural assets with physical assets, but they also resulted in a negative contribution to social assets. Unfortunately, the service delivered by ACs did not contribute to the access to human assets, nor to the access to financial assets or access to water for consumption from January to May. AC operations have been constrained by: (a) a lack of human and financial resources by which to sustain service delivery after the completion of NGO projects; (b) the willingness and commitment of AC committee members; and (c) the trust between AC committee members and smallholder farmers. (3) SEM predicts that adaptive capacity contributes to mitigating the impacts of drought and increasing participation in AC activities. While AC operations significantly impact adaptive capacity and mitigate the impact of droughts, participation in AC activities did not by itself contribute to AC operations, nor did it help with drought impacts. Through AC operations in developing countries such as Cambodia, smallholder farmers are empowered with opportunities and resources to promote agricultural development, income generation activities, and responses to the impacts of drought. AC operations provide awareness, skills, agricultural inputs, equipment, facilities, and markets, which contribute to smallholder farmers' human, financial, and social assets.

This paper was written with certain limitations, such as a lack of participation and observation during the election of AC committees and meetings. The analysis was primarily based on raw data, surveys, key informants, in-depth interviews, focus group discussions, and consultative meetings. In the future, researchers may wish to consider conducting fieldwork in more provinces for a nationwide survey; they may also wish to include other highly rice-productive regions, such as Battambang, Kampong Thom, Kandal, Prey Veng, and Svay Rieng, in their studies. This research fills a gap in the literature by increasing the understanding regarding the roles of AC operations in reducing drought impacts by increasing access to the five livelihood assets and in improving the adaptive capacity of smallholder farmers. Currently, AC operations are dependent upon the support of local governments and NGOs; additionally, ACs have faced difficulties in continuing to operate effectively after the NGO projects are completed. Further studies are recommended for researchers with respect to the capacity assessment of AC committees, the supporting mechanisms and local resources for AC operation, and for understanding the external dependency and sustainability of ACs.

Enhanced financial and technical capacity of ACs is required in the long term in order to promote the adaptive capacity of smallholder farmers to drought. ACs have roles in planning and delivering services, and they also help increase access to human and natural assets with support from local governments and NGOs. The private sector helps increase access to physical and financial assets. In the future, ACs should work closely with CoCs to empower smallholder farmers in terms of livelihood development and drought risk reduction. Simultaneously, NGOs and government agencies should continue building the capacity to generate sufficient revenues by which to operate activities for members alone. Actions implemented by ACs should be included as priorities within the annual budget under the commune investment plan. When the activity implementation of ACs is aligned with the priority of the commune investment plan, they can mobilize government agencies and NGOs for routine activities.

**Author Contributions:** Conceptualization, S.S. and N.C.; methodology, N.C., S.S., V.S. and P.N.; software, V.S. and P.N.; validation, S.S., P.N. and N.C.; formal analysis, V.S.; investigation, S.S.; resources, N.C.; data curation, N.C.; writing—original draft preparation, S.S. and N.C.; writing—review and editing, N.C., S.S. and P.N.; visualization, V.S. and P.N.; supervision, N.C.; project administration, N.C.; funding acquisition, N.C. All authors have read and agreed to the published version of the manuscript.

**Funding:** This research was carried out with the generous aid of a grant from the Ottawa-based International Development Resource Center (IDRC), Canada. The authors wish to acknowledge IDRC Canada for funding the project "Promoting Social Entrepreneurship in Disaster Risk Reduction to Building Community Resilience: Pilot in Malaysia and Cambodia (XX-2019-011)", which supported this study.

**Data Availability Statement:** The raw data regarding rainfall and supplementary irrigation between 2003 and 2020 were obtained from the Department of Meteorology, Ministry of Water Resources, and Meteorology (MoWRAM) and the Commune Database Online (CDB) of the Ministry of Planning (MoP), respectively.

**Acknowledgments:** The support of the Southeast Asia Disaster Prevention Research Initiative (SEADPRI-UKM), Universiti Kebangsaan, Malaysia, in this research is also appreciatively accredited. The authors wish to also thank the local authorities and officers of Pursat and Kampong Speu Provinces for their assistance with the fieldwork.

**Conflicts of Interest:** The authors declare no conflict of interest.

**Appendix A**

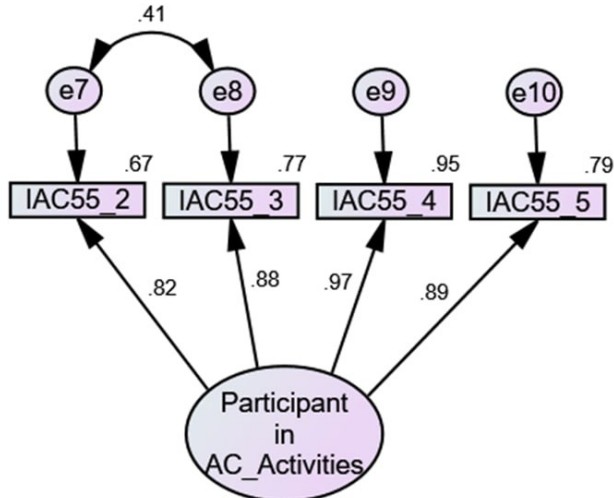

**Figure A1.** The first-order factor model.

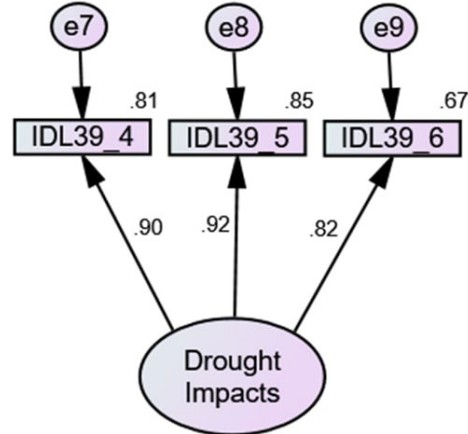

**Figure A2.** First-order factor model.

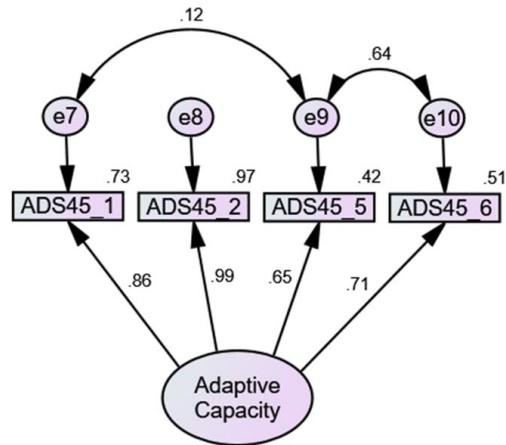

Model=Standardized estimates
Group=Group number 1
Ch-sqaure=.000,
df=0, Chi-square/df=\cmindf,
GFI=1.000 , AGFI=\AGFI,
NFI=\NFI, CFI=\CFI,
RMSEA=\RMSEA, P=\P

**Figure A3.** First-order factor model.

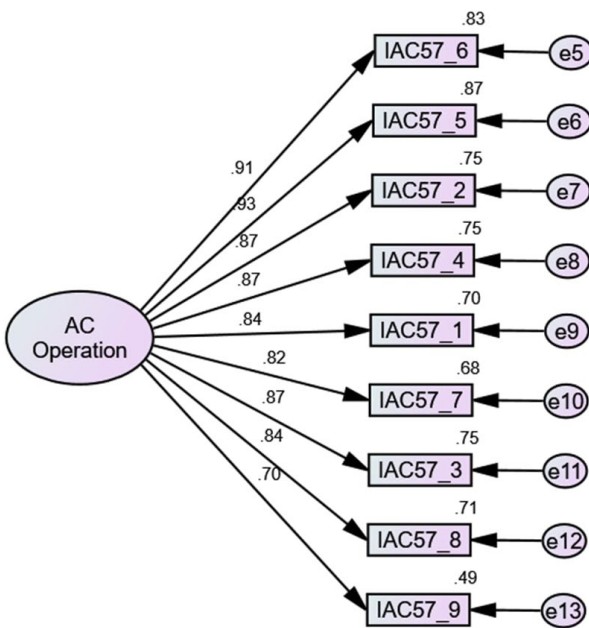

Model=Standardized estimates
Group=Group number 1
Ch-sqaure=37.078,
df=17, Chi-square/df=2.181,
GFI=.981 , AGFI=.950,
NFI=.991, CFI=.995,
RMSEA=.053, P=.003

**Figure A4.** First-order factor model.

**Table A1.** Structured Questionnaire of the Household survey.

| 39 | Please Rate the Impact of Drought on Your Socioeconomic Development. | | | | | | |
|---|---|---|---|---|---|---|---|
| | | N/A | Very Low | Low | Moderate | High | Very High |
| IDL-1 | Loss of employment | □0 | □1 | □2 | □3 | □4 | □5 |
| IDL-2 | Reduction in household income | □0 | □1 | □2 | □3 | □4 | □5 |
| IDL-3 | Reduction in spending on festivals | □0 | □1 | □2 | □3 | □4 | □5 |
| IDL-4 | Food scarcity | □0 | □1 | □2 | □3 | □4 | □5 |
| IDL-5 | Threatened household food security | □0 | □1 | □2 | □3 | □4 | □5 |
| IDL-6 | Limited food preference | □0 | □1 | □2 | □3 | □4 | □5 |
| IDL-7 | Conflicts for water in society | □0 | □1 | □2 | □3 | □4 | □5 |
| IDL-8 | Affected the schooling for children | □0 | □1 | □2 | □3 | □4 | □5 |
| IDL-9 | Population migration | □0 | □1 | □2 | □3 | □4 | □5 |
| IDL-10 | Health impact | □0 | □1 | □2 | □3 | □4 | □5 |
| IDL-11 | Homelessness and sense of loss | □0 | □1 | □2 | □3 | □4 | □5 |
| IDL-12 | Other (specify_____________) | □0 | □1 | □2 | □3 | □4 | □5 |
| 45 | To what degree were significant agricultural adaptation measures adopted by smallholder farmers | | | | | | |
| | | N/A | Very Low | Low | Moderate | High | Very High |
| | | □0 | □1 | □2 | □3 | □4 | □5 |
| ADS-1 | Changing crop calendar/cooperative dates | □0 | □1 | □2 | □3 | □4 | □5 |
| ADS-2 | Changing to low, water-consuming crops | □0 | □1 | □2 | □3 | □4 | □5 |
| ADS-3 | Keeping land unsown after the anticipated drought | □0 | □1 | □2 | □3 | □4 | □5 |
| ADS-4 | Changing traditional irrigation practices to sprinkler, drip irrigation | □0 | □1 | □2 | □3 | □4 | □5 |
| ADS-5 | Water harvesting (farm pond, in situ water conservation practice, etc.) | □0 | □1 | □2 | □3 | □4 | □5 |
| ADS-6 | Reducing wastage of water during the drought year | □0 | □1 | □2 | □3 | □4 | □5 |
| ADS-7 | Other (specify_____________) | □0 | □1 | □2 | □3 | □4 | □5 |
| 55 | To what degree are you involved in the following activities with the agricultural cooperative? | | | | | | |
| | | N/A | Very Low | Low | Moderate | High | Very High |
| | | □0 | □1 | □2 | □3 | □4 | □5 |
| IAC-1 | Participation in productive activities (planning and harvesting) | □0 | □1 | □2 | □3 | □4 | □5 |
| IAC-2 | A regular presence in meetings held by the farmer association. | □0 | □1 | □2 | □3 | □4 | □5 |
| IAC-3 | Participation in a training course organized by the farmer's association. | □0 | □1 | □2 | □3 | □4 | □5 |
| IAC-4 | Cooperation with the committees of the farmer's association. | □0 | □1 | □2 | □3 | □4 | □5 |
| IAC-5 | Participation in the decision-making of the farmer's associations about productive activities. | □0 | □1 | □2 | □3 | □4 | □5 |
| IAC57-6 | Other (specify_____________) | □0 | □1 | □2 | □3 | □4 | □5 |
| 56 | To what degree are you satisfied with the services delivered by the agricultural cooperative? | | | | | | |
| | | N/A | Very Low | Low | Moderate | High | Very High |
| | | □0 | □1 | □2 | □3 | □4 | □5 |
| IAC-1 | Adequate capital accumulation | □0 | □1 | □2 | □3 | □4 | □5 |
| IAC-2 | Availability of loans | □0 | □1 | □2 | □3 | □4 | □5 |
| IAC-3 | Management of loans | □0 | □1 | □2 | □3 | □4 | □5 |

**Table A1.** *Cont.*

| 39 | Please Rate the Impact of Drought on Your Socioeconomic Development. | | | | | | |
|---|---|---|---|---|---|---|---|
| | | N/A | Very Low | Low | Moderate | High | Very High |
| IAC-4 | Sufficiently skilled personnel | □0 | □1 | □2 | □3 | □4 | □5 |
| IAC-5 | Government support | □0 | □1 | □2 | □3 | □4 | □5 |
| IAC-6 | High literate level members | □0 | □1 | □2 | □3 | □4 | □5 |
| IAC-7 | Good management | □0 | □1 | □2 | □3 | □4 | □5 |
| IAC-8 | Sufficient business plans | □0 | □1 | □2 | □3 | □4 | □5 |
| IAC-9 | Large values of shares | □0 | □1 | □2 | □3 | □4 | □5 |
| IAC-10 | Sufficiency of access to credit facilities | □0 | □1 | □2 | □3 | □4 | □5 |
| IAC-11 | Access to competitive markets | □0 | □1 | □2 | □3 | □4 | □5 |
| IAC-12 | Other (specify________________) | □0 | □1 | □2 | □3 | □4 | □5 |
| 57 | To what degree are you satisfied with the implementation of agricultural cooperation? | | | | | | |
| | | N/A | Very Low | Low | Moderate | High | Very High |
| | | □0 | □1 | □2 | □3 | □4 | □5 |
| IAC57-1 | Providing capital and credit facilities | □0 | □1 | □2 | □3 | □4 | □5 |
| IAC57-2 | The availability of market services | □0 | □1 | □2 | □3 | □4 | □5 |
| IAC57-3 | Providing practical knowledge and the teaching of techniques | □0 | □1 | □2 | □3 | □4 | □5 |
| IAC57-4 | Receiving external support | □0 | □1 | □2 | □3 | □4 | □5 |
| IAC57-5 | Responding to members' needs | □0 | □1 | □2 | □3 | □4 | □5 |
| IAC57-6 | Knowing members' need | □0 | □1 | □2 | □3 | □4 | □5 |
| IAC57-7 | Having leadership and work capabilities | □0 | □1 | □2 | □3 | □4 | □5 |
| IAC57-8 | Having good bookkeeping/financial management | □0 | □1 | □2 | □3 | □4 | □5 |
| IAC57-9 | Enforcing internal regulation | □0 | □1 | □2 | □3 | □4 | □5 |
| IAC57-10 | Having good communications with local authorities | □0 | □1 | □2 | □3 | □4 | □5 |
| IAC57-11 | Other (specify________________) | □0 | □1 | □2 | □3 | □4 | □5 |

**Table A2.** List of Participants for Key Informants and In-depth Interview.

| Code | Institution | Date |
|---|---|---|
| Pers. Comm. K-1 | The Ministry of Agriculture, Forestry and Fisheries (MAFF) | August 2022 |
| Pers. Comm. K-2 | Heifer International Cambodia | May 2022 |
| Pers. Comm. K-3 | The Bakan district officer | December 2022 |
| Pers. Comm. K-5 | The Commune Councils (CoCs) in the Bakan district | December 2022 |
| Pers. Comm. K-5 | The District Office of Agriculture, Fishery and Forestry in Barsedth district | November 2022 |
| Pers. Comm. Interview-1 | The Ponlue Agricultural Cooperative in Bakan district | December 2022 |
| Pers. Comm. Interview-2 | The Chamreun Pal Agricultural Cooperative in Bakan district | December 2022 |
| Pers. Comm. Interview-3 | The Chamreoun Agricultural Cooperative | December 2022 |
| Pers. Comm. Interview-4 | The Agri-Productive Transport vehicle: a non-cold Chain Truck in Barsedth district | November 2022 |
| Pers. Comm. Interview-5 | The Chamrostean Agricultural Cooperative in Barsedth district | November 2022 |
| Consultive Meeting | The consultative meeting in the Bakan district | December 2022 |
| FGD 1 | The focus group discussion in the Bakan district | December 2022 |
| FGD 2 | The focus group discussion in the Barsedth district | November 2022 |

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
