# Peer review of "Roles of Agricultural Cooperatives (ACs) in Drought Risk Management among Smallholder Farmers in Pursat and Kampong Speu Provinces, Cambodia"

_water, doi:10.3390/w15081447_

Round 1
Reviewer 1 Report
This paper represents an interesting contribution for analyzing the roles of agricultural ooperatives (ACs) in drought risk management among smallholder famers in Cambodia. The introduction to the paper nicely suggests that there still are opportunities to research in this topic. However, in my opinion some further considerations are the following:
· I suggest considering a general, integrative theoretical approach to present the conceptual framework previously,and then write the paper from the angle of the specific chosen approach.
· Even though the paper is empirically oriented, good papers (either theoretical or empirical) always provide a review of both types of papers related to different disciplines of study. The reasons for this are that different types of readers may be interested in reading the paper and more importantly it helps to better evaluate the merits of thepaper’s contribution.
· The introduction should more clearly specify the novelty of the paper compared to other papers published in this area.
· The methodology needs justification in the literature review for a similar design.
· The last section (that is, Conclusion) is too brief and requires more elaboration in another section. Particularly, aConclusion section regarding the limitations and ways in which this research with differing intervention contributes to managerial and theoretical implications in the study is required. To the end of the paper, I suggest including several subsections renaming this section as follows Conclusions, Limitations, and Implications.
· In my opinion, results are more indicative rather than representative. More limitations and managerialcontributions of the study in terms of the generalization of the findings should be added.
Author Response
Dear Reviewer
Thank for your kind comments and advise. We have addressed them carefully with our best knowledge.
With best regards
Serey

Reviewer 2 Report
Sloppy description of a primarily descriptive piece decorated by much pseudo-analysis. You contibute nothing to better understanding of risk management strategies in general or in this particular geographical case study. Some of the remarks about the better design of ACs might be useful observations but there is no real contribution to a better understanding of the role of coperatives. Some of the many problems are marked in the attached MS

Author Response

(The authors gave the same response as above.)

Reviewer 3 Report
The manuscript is well-written but needs some minor revisions as follows:
1. Please clarify the determination of sample size.
2. Please provide the questionnaire used for data collection as supplementary material.
3. The results are satisfactory, but the discussion should come with a more detailed comparison with similar studies.
Author Response

(The authors gave the same response as above.)

Round 2
Reviewer 1 Report
The authors have made some improvements compared to the previous submission. In my opinion, there are some recommendations as follows:
1. In terms of explaining the paper contribution from a theoretical point of view, I am not completely satisfied. In section 1, the problem description is described and presented, but it seems reasonable to develop some arguments in the end of this first section, before Section 2. Conceptual Framework.
- The hypothesis needs reformulation in the literature review for a similar design. In my opinion, authors must reformulate the hypothesis in a precise, testable statement of what the researchers predict will be the outcome of the study.
- It is recommended to rename the conclusion section as follows: Conclusions, Limitations and Further Research, and Implications. That is, in the last section, Conclusion, it is also recommended to briefly present subsections and describe the findings, recommendations, and the main discussion, even for readers who are not familiar with the topic.
- In my opinion, results are more indicative rather than representative. More limitations and managerial contributions of the study in terms of the generalization of the findings should be added.
Author Response
Dear Reviewer
Thank for your kind advice and suggestion to improve our paper quality. We have worked to to address your comments seriously.
With best regards
Serey
